# The prevalence of and demographic factors associated with radiographic knee osteoarthritis in Korean adults aged ≥ 50 years: The 2010–2013 Korea National Health and Nutrition Examination Survey

**Jae Won Hong, Jung Hyun Noh, Dong-Jun Kim**[ID]*

Department of Internal Medicine, Ilsan-Paik Hospital, College of Medicine, Inje University, Koyang, Gyeonggi-do, Republic of Korea

* djkim@paik.ac.kr

## Abstract

### Background

To reduce the social burden of knee osteoarthritis (OA) by addressing it in the early stages in the population at greatest risk, the epidemiology of knee OA needs to be understood and associated demographic factors need to be identified.

### Objectives

We evaluated the weighted prevalence of and demographic factors associated with radiographic knee OA in Korean adults.

### Methods

We analyzed data from 12,287 individuals aged ≥ 50 years who had radiographs of the knee taken in the 2010–2013 Korea National Health and Nutrition Examination Survey (KNHANES). Radiographic knee OA was defined based on the Kellgren–Lawrence grade, as follows: 0: No abnormal finding 1: Mild degenerative changes, minute osteophytes 2: Mild knee OA, definite osteophytes 3: Moderate knee OA, moderate joint-space narrowing and definite osteophytes 4: Severe knee OA, severe joint-space narrowing with subchondral sclerosis.

### Results

We found that the prevalence of radiographic knee OA in the Korean adult population was 35.1%. Logistic regression analyses were performed to identify factors associated independently with radiographic knee OA, with age, sex, area of residence, education level, household income, and obesity serving as covariates. Women were at greater risk than men of having knee OA (OR 2.12, 95% CI 1.90–2.37, p < 0.001). Compared with subjects aged 50–59 years, adults aged ≥ 80 years were at 8.87-fold (95% CI 7.12–11.06, p < 0.001) greater

**Data Availability Statement:** Interested researchers can access raw data from the Korean CDC website by signing up for membership

(https://knhanes.cdc.go.kr/knhanes/index.do). On the blue bar at the top of the website, click the third menu item, "원시자료." Then select the second submenu below the blue bar, "시자료 다운로드," to enter your email address and download the raw data from 1998-2016 Korea National Health and Nutrition Examination Survey database using SAS or SPSS.

**Funding:** The author(s) received no specific funding for this work.

**Competing interests:** The authors have declared that no competing interests exist.

risk of having knee OA. Residence in a rural area was associated with a greater risk of having radiographic knee OA than was residence in an urban area (OR 1.26, 95% CI 1.08–1.48, p = 0.004), regardless of knee OA severity (Kellgren–Lawrence grades ≥2, ≥3, and 4). Elementary school graduates had 1.71-fold (p < 0.001) greater risks of having knee OA than did college graduates. Household incomes ≤24th percentile were associated with a greater risk of having knee OA compared with those ≥75th percentile (OR 1.28, 95% CI 1.08–1.52, p = 0.004). Obesity was associated with an approximately two-fold greater risk of knee OA, regardless of knee OA severity (Kellgren–Lawrence grades ≥2, ≥3, and 4).

## Conclusions

Using data from the 2010–2013 KNHANES and defining knee OA as Kellgren–Lawrence grade ≥ 2, we found that the prevalence of radiographic knee OA was 35.1% (24.4% in men, 44.3% in women) in a representative sample of Korean adults aged ≥ 50 years, with the highest prevalence (78.7%) observed in women aged ≥ 80 years. Low socioeconomic status and traditional factors, including age, female sex, and obesity, were associated with the risk of knee OA.

## Introduction

Osteoarthritis (OA) of the knee is a complex peripheral joint disorder with multiple risk factors that results in progressive loss of function, pain, and stiffness [1]. Frequent knee pain affects approximately 25% of adults, and OA is the most common cause of knee pain in people older than 50 years [2,3].

Clinically, knee OA consists of joint symptoms and evidence of structural change, usually demonstrated radiographically [3]. According to the European League Against Rheumatism's recommendations, as no guideline for the clinical diagnosis of knee OA is currently available, plain radiography is often used as the gold standard for the assessment of knees with clinical evidence of OA [4]. Most studies have involved radiographic assessment as the primary means of identifying OA, with the Kellgren–Lawrence scale used to grade OA based on the definite presence of osteophytes [5]. The presence of osteophytes in knee OA correlates well with its symptoms [6].

Given that aging and obesity are major risk factors for knee OA, this disorder and total knee replacement have become substantially more common in recent decades [7,8]. According to the 2010 Global Burden of Diseases study, the burden of OA is increasing, most rapidly among musculoskeletal disorders in terms of disability-adjusted life years; it will impose new challenges on health systems, along with mental disorders and diabetes [9]. To reduce the social burden of knee OA by addressing it in the early stages in the population at greatest risk, the epidemiology of knee OA needs to be understood and associated demographic factors need to be identified.

In this study, we investigated the prevalence of and demographic factors associated with radiographic knee OA based on the Kellgren–Lawrence grade in Korean adults aged ≥ 50 years using data from the 2010–2013 Korea National Health and Nutrition Examination Survey (KNHANES).

## Methods

### Study population and data collection

This study was based on data from the 2010–2013 KNHANES, a cross-sectional, nationally representative survey conducted by the Korean Center for Disease Control for Health Statistics. As

described in detail previously [10,11], the KNHANES is independent dataset obtained from the general population of Korea, similar to data from the National Health and Nutrition Examination Survey (NHANES) in the United States. The KNHANES has been conducted periodically since 1998 to assess the health and nutritional status of the civilian, noninstitutionalized population of Korea. Participants are selected using proportional-allocation systematic sampling with multistage stratification. Standardized interviews are conducted in the homes of the participants to collect information on demographic variables, family and medical histories, medications used, and various other health-related variables. The interviewers use an established questionnaire to record the demographic and socioeconomic characteristics of the subjects, including age, education level, occupation, household income, marital status, smoking status, alcohol consumption, exercise habits, previous and current diseases, and family disease history.

Of the 33,552 participants in the 2010–2013 KNHANES, data from 12,287 individuals aged ≥ 50 years who had radiographs of the knee taken were analyzed in this study.

## Assessment of radiographic knee OA

Bilateral anteroposterior and lateral plain radiographs of the knees were taken using a DigiRAD-PG 9M. Two radiologists performed OA examinations and independent assessments by webhard uploading and downloading using the Kellgren–Lawrence grading system. For differences of one grade between radiologists, the higher grade was accepted. For discrepancies exceeding one grade, a third radiologist was consulted, and the grade concordant with the third assessment was accepted. Radiographic OA grading agreement rates for 2010–2013 were 87.96%, 95.18%, 89.62%, and 85.19%, respectively. Weighted kappa coefficients for inter-rater reliability in 2010–2013 were 0.6522, 0.7407, 0.8383, and 0.6842, respectively, indicating fair to very high degrees of agreement.

Radiographic knee OA was defined based on the Kellgren–Lawrence grade, as follows:

0: No abnormal finding

1: Mild degenerative changes, minute osteophytes

2: Mild knee OA, definite osteophytes

3: Moderate knee OA, moderate joint-space narrowing and definite osteophytes

4: Severe knee OA, severe joint-space narrowing with subchondral sclerosis

## Ethical considerations

The institutional review board of Ilsan Paik Hospital, Republic of Korea, approved this study. After the study proposal had been approved, the KNHANES dataset was made available at the request of the investigator. The study was exempt from the requirement for consent because the dataset did not include personal information and KNHANES participants had already given consent.

## Statistical analyses

The KNHANES participants were not sampled randomly. The survey was designed using a complex, stratified, multistage probability-sampling model; consequently, individual participants were not equally representative of the Korean population. To obtain representative prevalence rates from the dataset, consideration of the power of each participant (sample weight) as a representative of the Korean population was necessary. Following approval from the

Korea Centers for Disease Control and Prevention, we received a survey dataset that included information on the survey location, age, sex, and various other factors and the sample weight for each participant. The survey sample weights, which were calculated using the sampling and response rates and age/sex proportions of the reference population (2005 Korean National Census Registry), were used in all of the analyses to produce representative estimates of the noninstitutionalized Korean civilian population. The statistical analyses were performed using SPSS ver. 21.0 for Windows (SPSS, Chicago, IL, USA). To compare the weighted prevalence of radiographic knee OA by sociodemographic factors, chi-squared tests and general linear model were performed. The prevalence of radiographic knee OA was analyzed using age (50–59, 60–69, 70–79, ≥80 years old), sex (men/women), area of residence (urban [Dong]) /rural [Eup/Myeon]), education level (elementary school/junior high school/senior high school/college graduated), number of family members (1/2/3/≥4), household income (≤ 24$^{th}$, 25-49$^{th}$, 50-74$^{th}$,≥75$^{th}$ percentile), occupation (managers and professionals/clerical support workers/ service and sales workers/skilled agricultural, forestry and fishery workers/craft, plant, or machine operators and assemblers/laborers/unemployed (including students and house wives), and obesity [body mass index (BMI) ≥ 25 kg/m$^2$] as covariates. Logistic regression analyses were used to calculate the odds ratio (OR) for radiographic knee OA with age(50–59, 60–69, 70–79, ≥80 years old), sex(men/women), area of residence(urban/rural), education level(elementary school/junior high school/senior high school/college graduated), household income(≤ 24$^{th}$, 25-49$^{th}$, 50-74$^{th}$,≥75$^{th}$ percentile), and obesity(no/yes) serving as covariates. All tests were two sided and $p < 0.05$ was considered to be indicative of statistical significance.

## Results

### Weighted demographic and clinical characteristics of the study population

The weighted demographic and clinical characteristics of the study population are shown in Table 1 shows the weighted demographic and clinical characteristics of the study population. The mean weighted age was 62.5 years [95% confidence interval (CI) 62.2–62.7], and 54% of the participants were female. The weighted percentage of obesity (BMI ≥ 25 kg/m$^2$) was 35.2%. The weighted percentages of Kellgren–Lawrence grades 0–4 were 40.8%, 24.0%, 14.3%, 14.6%, and 6.2%, respectively. Based on Kellgren–Lawrence grade ≥ 2, we found that the prevalence of radiographic knee OA in the Korean adult population was 35.1%.

### Weighted prevalence of radiographic knee OA according to age and sex

The weighted prevalences of Kellgren–Lawrence grades ≥2, ≥3, and 4 in the study population were 35.1% (33.7–36.6%), 20.8% (19.8–22.0%), and 6.2% (5.7–6.8%), respectively (Table 2). The weighted prevalence of radiographic knee OA increased with age, irrespective of knee OA severity (Kellgren–Lawrence grades ≥2, ≥3, and 4). The overall weighted prevalence of radiographic knee OA in adults aged ≥ 80 years was 71.6% (67.6–75.3%).

In men, the weighted prevalences of Kellgren–Lawrence grades ≥2, ≥3, and 4 were 24.4% (22.7–26.2%), 10.2% (9.2–11.3%), and 2.1% (1.7–2.6%), respectively. The weighted prevalence of radiographic knee OA in men increased with age, irrespective of knee OA severity (Kellgren–Lawrence grades ≥2, ≥3, and 4). The weighted prevalence of radiographic knee OA in men aged ≥ 80 years was 55.5% (57.5–63.1%).

In women, the weighted prevalences of Kellgren–Lawrence grades ≥2, ≥3, and 4 were 44.3% (42.7–46.0%), 29.9% (28.4–31.5%), and 9.7% (8.9–10.6%), respectively. The weighted prevalence of radiographic knee OA was higher in women than in men, irrespective of knee OA severity (Kellgren–Lawrence grades ≥2, ≥3, and 4). The weighted prevalence of radiographic knee OA in women aged ≥ 80 years was 78.7% (74.5–82.4%).

**Table 1. Demographic and clinical characteristics of the study population.**

| Variables | Unweighted Number (%) | Weighted Number (%) |
|---|---|---|
| Total | 12,287 | 14,837,279 |
| Sex | | |
| Men | 5,231 (42.6) | 6,846,055 (46.1) |
| Women | 7,056 (57.4) | 7,991,224 (53.9) |
| Age (years) | | |
| 50–59 | 4,513 (36.7) | 7,006,966 (47.2) |
| 60–69 | 3,967 (32.3) | 4,079,939 (27.5) |
| 70–79 | 3,075 (25.0) | 2,922,903 (19.7) |
| 80- | 732 (6.0) | 827,471 (5.6) |
| Area of residence | | |
| Urban | 8,966 (73.0) | 10,889,741 (73.4) |
| Rural | 3,321 (27.0) | 3,947,538 (26.6) |
| Education | | |
| Elementary school graduated | 6,093 (49.6) | 7,048,684 (47.5) |
| Junior high school graduated | 2,027 (16.5) | 2,629,140 (17.7) |
| Senior high school graduated | 2,793 (22.7) | 3,534,363 (23.8) |
| College graduated | 1,374 (11.2) | 1,625,093 (11.0) |
| Family member (n) | | |
| 1 | 1,497 (12.2) | 1,561,237 (10.5) |
| 2 | 5,132 (41.8) | 5,455,879 (36.8) |
| 3 | 2,841 (23.1) | 3,852,005 (26.0) |
| $\geq 4$ | 2,817 (22.9) | 3,968,122 (26.7) |
| Household income | | |
| $\leq 24^{th}$ percentile | 4,048 (32.9) | 4,511,878 (30.4) |
| 25-49$^{th}$ percentile | 3,132 (25.5) | 3,748,286 (25.3) |
| 50-74$^{th}$ percentile | 2,484 (20.2) | 3,181,774 (21.4) |
| $\geq 75^{th}$ percentile | 2,623 (21.3) | 3,395,841 (22.9) |
| Occupation | | |
| Managers and professionals | 632 (5.1) | 845,042 (5.7) |
| Clerical support workers | 321 (2.6) | 451,428 (3.0) |
| Service and sales workers | 1,176 (9.6) | 1,665,965 (11.2) |
| Skilled agricultural, forestry and fishery workers | 1,496 (12.2) | 1,661,755 (11.2) |
| Craft, plant, or machine operators and assemblers | 960 (7.8) | 1,542,072 (10.4) |
| Laborers | 1,373 (11.2) | 1,674,336 (11.3) |
| Unemployed(including students and house wives) | 6,329 (51.5) | 6,996,681 (47.2) |
| Obesity (BMI $\geq 25kg/m^2$) | 4,269 (34.7) | 5,225,644 (35.2) |
| Kellgren-Lawrence grade | | |
| 0 | 4,690 (38.2) | 6,060,519 (40.8) |
| 1 | 2,944 (24.0) | 3,563,758 (24.0) |
| 2 (mild) | 1,848 (15.0) | 2,121,220 (14.3) |
| 3 (moderate) | 1,929 (15.7) | 2,167,619 (14.6) |
| 4 (severe) | 876 (7.1) | 924,165 (6.2) |

## Weighted prevalence of radiographic knee OA according to demographic and clinical characteristics

Table 3 shows the weighted unadjusted and adjusted prevalences of radiographic knee OA according to demographic and clinical variables after adjustment for age, sex, area of

**Table 2. Weighted prevalence of radiographic knee osteoarthritis in the Korean adults (≥ 50 years).**

| | Number (unweighted/ weighted) | Prevalence of mild to severe knee osteoarthritis (Kellgren-Lawrence grade ≥ 2) | Prevalence of moderate to severe knee osteoarthritis (Kellgren-Lawrence grade ≥3) | Prevalence of severe knee osteoarthritis (Kellgren-Lawrence grade = 4) |
|---|---|---|---|---|
| **Total** | | | | |
| Total | 12,287/14,837,279 | 35.1 (33.7–36.6) | 20.8 (19.8–22.0) | 6.2 (5.7–6.8) |
| 50–59 years old | 4,513/7,006,966 | 18.9 (17.4–20.6) | 8.3 (7.3–9.4) | 1.0 (0.8–1.4) |
| 60–69 years old | 3,967/4,079,939 | 39.6 (37.4–41.8) | 22.0 (20.3–23.8) | 5.5 (4.7–6.5) |
| 70–79 years old | 3,075/2,922,903 | 57.5 (55.2–59.7) | 39.2 (36.9–41.4) | 13.8 (12.2–15.5) |
| ≥ 80 years old | 732/827,471 | 71.6 (67.6–75.3) | 56.4 (52.0–60.7) | 26.9 (23.1–31.1) |
| **Men** | | | | |
| Total | 5,231/6,846,055 | 24.4 (22.7–26.2) | 10.2 (9.2–11.3) | 2.1 (1.7–2.6) |
| 50–59 years old | 1,872/3,467,680 | 14.3 (12.4–16.4) | 4.8 (3.8–6.2) | 0.6 (0.3–1.1) |
| 60–69 years old | 1,780/1,934,143 | 27.8 (25.1–30.7) | 10.5 (8.8–12.5) | 1.8 (1.2–2.7) |
| 70–79 years old | 1,326/1,191,503 | 41.8 (38.4–45.3) | 20.5 (18.0–23.3) | 5.4 (4.2–7.0) |
| ≥ 80 years old | 253/252,729 | 55.5 (57.5–63.1) | 33.9 (26.9–41.7) | 10.7 (6.3–17.5) |
| **Women** | | | | |
| Total | 7,056/7,991,244 | 44.3 (42.7–46.0) | 29.9 (28.4–31.5) | 9.7 (8.9–10.6) |
| 50–59 years old | 2,641/3,539,286 | 23.5 (21.4–25.7) | 11.7 (10.2–13.4) | 1.5 (1.1–2.0) |
| 60–69 years old | 2,187/2,145,796 | 50.2 (47.5–52.8) | 32.4 (29.9–35.0) | 8.9 (7.6–10.5) |
| 70–79 years old | 1,749/1,731,400 | 68.2 (65.6–70.8) | 52.0 (49.1–54.9) | 19.6 (17.3–22.1) |
| ≥ 80 years old | 479/574,743 | 78.7 (74.5–82.4) | 66.3 (61.3–71.0) | 34.0 (29.2–39.2) |

Data are expressed as mean (95% CI)

residence, education level, number of family members, household income, occupation, and obesity.

The unadjusted and adjusted weighted prevalences of radiographic knee OA were lower in urban areas than in rural areas [adjusted, 34.0% (32.6–35.5%) vs. 38.2% (35.5–40.9%), p = 0.008].

Education level was correlated inversely with the prevalence of radiographic knee OA, before and after adjustment. Elementary school graduates had a higher prevalence of radiographic knee OA than did college graduates [adjusted, 37.6% (35.7–39.5%) vs. 29.5% (26.7–32.3%), p < 0.001].

The number of family members was associated negatively with the unadjusted prevalence of radiographic knee OA (p < 0.001). However, after adjustment, the statistical significance did not persist.

Household income was also associated negatively with the prevalence of radiographic knee OA before and after adjustment. Subjects with household incomes ≤24th percentile had

**Table 3. Weighted prevalence of radiographic knee osteoarthritis (Kellgren-Lawrence grade ≥ 2) according to the demographic and clinical factors.**

| Variables | | Unadjusted | | Adjusted for age, sex, | | Adjusted for age, sex, and other variables* | |
|---|---|---|---|---|---|---|---|
| Area of residence | Urban | 32.4 (30.8–33.9) | Reference | 33.7 (32.2–35.2) | Reference | 34.0 (32.6–35.5) | Reference |
| | Rural | 42.8 (39.9–45.8) | <0.001 | 39.1 (36.3–41.9) | **0.001** | 38.2 (35.5–40.9) | **0.008** |
| Education | | | <0.001 | | **<0.001** | | **<0.001** |
| | Elementary school graduated | 47.5 (45.6–49.5) | Reference | 38.4 (36.5–40.3) | Reference | 37.6 (35.7–39.5) | Reference |
| | Junior high school graduated | 28.9 (26.3–31.4) | <0.001 | 34.4 (31.9–37.0) | 0.007 | 34.3 (31.7–36.8) | 0.020 |
| | Senior high school graduated | 23.5 (21.5–25.3) | <0.001 | 32.4 (30.4–34.4) | <0.001 | 33.4 (31.4–35.5) | 0.002 |
| | College graduated | 16.9 (14.2–19.6) | <0.001 | 28.0 (25.3–30.8) | <0.001 | 29.5 (26.7–32.3) | <0.001 |
| Family member (n) | | | <0.001 | | 0.200 | | 0.855 |
| | 1 | 51.0 (47.6–54.4) | Reference | 36.7 (33.7–39.6) | Reference | 35.6 (32.7–38.6) | Reference |
| | 2 | 39.6 (37.7–41.6) | <0.001 | 36.1 (34.3–37.9) | 0.718 | 35.5 (33.7–37.3) | 0.941 |
| | 3 | 28.5 (26.2–30.8) | <0.001 | 33.6 (31.5–35.8) | 0.099 | 34.4 (32.3–36.5) | 0.527 |
| | ≥ 4 | 29.2 (26.8–31.5) | <0.001 | 34.7 (32.5–36.9) | 0.293 | 35.2 (33.0–37.3) | 0.806 |
| Household income | | | <0.001 | | **<0.001** | | **0.010** |
| | ≤ 24th percentile | 50.0 (47.9–52.1) | Reference | 38.8 (36.8–40.9) | Reference | 37.6 (35.6–39.7) | Reference |
| | 25-49th percentile | 33.2 (31.0–35.3) | <0.001 | 33.7 (31.7–35.7) | <0.001 | 33.4 (31.4–35.4) | 0.002 |
| | 50-74th percentile | 29.3 (26.8–31.8) | <0.001 | 35.2 (32.7–37.6) | 0.016 | 35.5 (33.0–37.9) | 0.168 |
| | ≥ 75th percentile | 23.0 (20.7–25.3) | <0.001 | 31.8 (29.5–34.1) | <0.001 | 33.4 (31.1–35.7) | 0.007 |
| Occupation | | | <0.001 | | **0.001** | | 0.192 |
| | Managers and professionals | 14.7 (11.4–17.9) | Reference | 31.4 (28.1–34.7) | Reference | 33.8 (30.2–37.3) | Reference |
| | Clerical support workers | 17.4 (12.3–22.6) | 0.357 | 35.1 (30.0–40.2) | 0.203 | 37.6 (32.6–42.6) | 0.182 |
| | Service and sales workers | 25.1 (22.1–28.1) | <0.001 | 33.7 (30.8–36.5) | 0.290 | 33.9 (31.2–36.7) | 0.942 |
| | Skilled agricultural, forestry and fishery workers | 39.0 (35.1–42.8) | <0.001 | 40.4 (36.8–44.1) | <0.001 | 37.8 (34.4–41.3) | 0.112 |
| | Craft, plant, or machine operators and assemblers | 20.3 (17.1–23.6) | 0.016 | 38.2 (35.0–41.5) | 0.002 | 37.7 (34.4–40.9) | 0.104 |
| | Laborers | 36.3 (33.2–39.5) | <0.001 | 35.7 (32.6–38.8) | 0.065 | 35.5 (32.5–38.5) | 0.464 |
| | Unemployed(including students and house wives) | 43.2 (41.5–45.0) | <0.001 | 33.9 (32.1–35.6) | 0.203 | 34.1 (32.4–35.9) | 0.853 |
| Obesity (BMI ≥ 25kg/m²) | No | 30.2 (28.7–31.8) | Reference | 30.0 (28.6–31.4) | Reference | 30.1 (28.6–31.5) | Reference |
| | Yes | 44.2 (42.1–46.3) | <0.001 | 44.6 (42.7–46.5) | **<0.001** | 44.5 (42.6–46.4) | **<0.001** |

Data are expressed as mean (95% CI) *Other variables include area of residence, education level, number of family members, household income, occupation, and obesity.

higher unadjusted and adjusted prevalences of knee OA than did subjects with household incomes in the 25–49th percentiles, 50–74th percentiles, and ≥75th percentile (p < 0.001).

Regarding occupation, with managers and professionals serving as controls, service and sales workers (p < 0.001); skilled agricultural, forestry, and fishery workers (p < 0.001); assemblers (p = 0.016); laborers (p<0.001); and unemployed subjects (p<0.001) had higher prevalences of knee OA. After adjustment for age and sex, with managers and professionals serving as controls, only skilled agricultural, forestry, and fishery workers (p < 0.001) and assemblers (p = 0.002) had a higher prevalences of knee OA. However, the statistical significance did not persist after adjustment for age, sex, area of residence, education level, number of family members, household income, and obesity.

Obesity was associated positively with a higher prevalence of radiographic knee OA, before and after adjustment (p < 0.001).

## Logistic regression analyses of radiographic knee OA

Logistic regression analyses were performed to identify factors associated independently with radiographic knee OA, with age, sex, area of residence, education level, household income, and obesity serving as covariates (Table 4).

Women were at greater risk than men of having knee OA (Kellgren–Lawrence grade $\geq$ 2; OR 2.12, 95% CI 1.90–2.37, p < 0.001). Women were at 3.30-fold greater risk of having severe knee OA (Kellgren–Lawrence grade = 4) than were men (p < 0.001).

Compared with subjects aged 50–59 years, adults aged $\geq$ 80 years were at 8.87-fold (95% CI 7.12–11.06, p < 0.001) greater risk of having knee OA (Kellgren–Lawrence grade $\geq$ 2). For severe knee OA (Kellgren–Lawrence grade = 4), the risk was more than 20 times (95% CI 14.53–30.97, p < 0.001) greater among adults aged $\geq$ 80 years than among those aged 50–59 years.

Residence in a rural area was associated with a greater risk of having radiographic knee OA than was residence in an urban area (OR 1.26, 95% CI 1.08–1.48, p = 0.004), regardless of knee OA severity (Kellgren–Lawrence grades $\geq$2, $\geq$3, and 4).

Elementary school graduates had 1.71-fold (p < 0.001) and 2.49-fold (p = 0.006) greater risks of having knee OA (Kellgren–Lawrence grade $\geq$ 2) and severe knee OA (Kellgren–Lawrence grade = 4), respectively, than did college graduates.

Household incomes $\leq$24th percentile were associated with a greater risk of having knee OA compared with those $\geq$75th percentile (OR 1.28, 95% CI 1.08–1.52, p = 0.004).

Obesity was associated with an approximately two-fold greater risk of knee OA, regardless of knee OA severity (Kellgren–Lawrence grades $\geq$2, $\geq$3, and 4).

**Table 4. Logistic regression analyses for radiographic knee osteoarthritis.**

| Variables | | Kellgren-Lawrence grade $\geq$ 2 | | Kellgren-Lawrence grade $\geq$ 3 | | Kellgren-Lawrence grade = 4 | |
|---|---|---|---|---|---|---|---|
| | | Odd ratio (95% CI) | P | Odd ratio (95% CI) | P | Odd ratio (95% CI) | P |
| Sex | | | | | | | |
| | Men | Reference | | Reference | | Reference | |
| | Women | 2.12 (1.90–2.37) | <0.001 | 3.10 (2.72–3.54) | <0.001 | 3.30 (2.67–4.08) | <0.001 |
| Age (years) | 50–59 | Reference | <0.001 | Reference | <0.001 | Reference | <0.001 |
| | 60–69 | 2.54 (2.23–2.90) | <0.001 | 2.60 (2.21–3.07) | <0.001 | 4.20 (2.97–5.95) | <0.001 |
| | 70–79 | 4.90 (4.25–5.64) | <0.001 | 5.47 (4.57–6.54) | <0.001 | 9.70 (6.74–13.97) | <0.001 |
| | 80 | 8.87 (7.12–11.06) | <0.001 | 10.81 (8.59–13.61) | <0.001 | 21.21 (14.53–30.97) | <0.001 |
| Area of residence | Urban | Reference | | Reference | | Reference | |
| | Rural | 1.26 (1.08–1.48) | 0.004 | 1.31 (1.11–1.54) | 0.001 | 1.35 (1.09–1.67) | 0.005 |
| Education | | | | | | | |
| | College graduated | Reference | <0.001 | Reference | <0.001 | Reference | <0.001 |
| | Senior high school graduated | 1.32 (1.05–1.66) | 0.016 | 1.21 (0.89–1.66) | 0.229 | 1.08 (0.52–2.21) | 0.842 |
| | Junior high school graduated | 1.45 (1.14–1.85) | 0.003 | 1.37 (1.00–1.89) | 0.053 | 1.83 (0.92–3.65) | 0.086 |
| | Elementary school graduated | 1.71 (1.36–2.14) | <0.001 | 1.84 (1.36–2.49) | <0.001 | 2.49 (1.30–4.76) | 0.006 |
| Household income | | | | | | | |
| | $\geq$ 75th percentile | Reference | 0.004 | Reference | 0.015 | Reference | 0.026 |
| | 50-74th percentile | 1.15 (0.96–1.36) | 0.121 | 1.07 (0.87–1.32) | 0.527 | 0.83 (0.59–1.18) | 0.306 |
| | 25-49th percentile | 1.03 (0.87–1.22) | 0.753 | 1.21 (0.97–1.50) | 0.091 | 0.88 (0.64–1.20) | 0.419 |
| | $\leq$ 24th percentile | 1.28 (1.08–1.52) | 0.004 | 1.36 (1.11–1.66) | 0.003 | 1.17 (0.87–1.58) | 0.299 |
| Obesity (BMI $\geq$ 25kg/m$^2$) | | | | | | | |
| | No | Reference | | Reference | | Reference | |
| | Yes | 2.10 (1.89–2.34) | <0.001 | 2.21 (1.97–2.48) | <0.001 | 2.12 (1.78–2.51) | <0.001 |

Additionally, logistic regression analyses were performed to identify factors associated independently with radiographic knee OA according to sex with age, area of residence, education level, household income, and obesity serving as covariates (Table 5).

In men, age, education level, and obesity were associated with radiographic knee OA. However, area of residence and household income were not.

In women, age, area of residence, education level, household income, and obesity were all associated with radiographic knee OA

## Discussion

Using data from the 2010–2013 KNHANES, we found that the prevalence of radiographic knee OA was 35.1% (24.4% in men, 44.3% in women) in a representative sample of Korean adults aged $\geq$ 50 years, with the highest prevalence (78.7%) observed in women aged $\geq$ 80 years.

Based on the NHANES III, the prevalences of radiographic knee OA and symptomatic knee OA were 37.4% and 12.1%, respectively, among adults aged > 60 years [12]. In the 2002–2005 Framingham Osteoarthritis Study, the age- and BMI-adjusted prevalences of radiographic knee OA in women and men were 35.4% and 35.1%, respectively [7].

Although we could not directly compare the prevalences of radiographic knee OA in general populations among countries because of the use of different data collection and analysis methodologies, the prevalence of radiographic knee OA (Kellgren–Lawrence grade $\geq$ 2) in South Korea seems to be similar to the global estimate.

Our data also suggest that sociodemographic factors, such as low education level and low household income, are associated with the risk of radiographic knee OA, in addition to the traditional factors of age, female sex, and obesity. However, the number of family members and

**Table 5. Logistic regression analyses for radiographic knee osteoarthritis (Kellgren-Lawrence grade $\geq$ 2) according to sex.**

| | | Men | | Women | |
|---|---|---|---|---|---|
| **Variables** | | **Odd ratio (95% CI)** | **P** | **Odd ratio (95% CI)** | **P** |
| Age (years) | 50–59 | Reference | <0.001 | Reference | <0.001 |
| | 60–69 | 2.20 (1.80–2.69) | <0.001 | 2.79 (2.36–3.30) | <0.001 |
| | 70–79 | 4.08 (3.26–5.10) | <0.001 | 5.58 (4.64–6.70) | <0.001 |
| | 80 | 7.09 (4.90–10.25) | <0.001 | 10.33 (7.82–13.65) | <0.001 |
| Area of residence | Urban | Reference | | Reference | |
| | Rural | 1.16 (0.92–1.47) | 0.207 | 1.33 (1.13–1.58) | 0.001 |
| Education | | | | | |
| College graduated | | Reference | 0.002 | Reference | 0.008 |
| Senior high school graduated | | 1.42 (1.08–1.86) | 0.012 | 1.23 (0.87–1.73) | 0.249 |
| Junior high school graduated | | 1.51 (1.11–2.06) | 0.008 | 1.40 (0.97–2.01) | 0.072 |
| Elementary school graduated | | 1.74 (1.31–2.32) | <0.001 | 1.59 (1.14–2.23) | 0.007 |
| Household income | | | | | |
| $\geq 75^{th}$ percentile | | Reference | 0.758 | Reference | 0.001 |
| 50-74$^{th}$ percentile | | 1.09 (0.84–1.41) | 0.536 | 1.19 (0.97–1.48) | 0.101 |
| 25-49$^{th}$ percentile | | 1.03 (0.80–1.34) | 0.797 | 1.04 (0.84–1.28) | 0.755 |
| $\leq 24^{th}$ percentile | | 1.14 (0.87–1.49) | 0.351 | 1.39 (1.14–1.70) | 0.001 |
| Obesity (BMI $\geq$ 25kg/m$^2$) | | | | | |
| No | | Reference | | Reference | |
| Yes | | 1.80 (1.52–2.14) | <0.001 | 2.30 (2.00–2.65) | <0.001 |

occupation were not associated with radiographic knee OA after adjusting for age, sex, area of residence, education level, household income, and obesity.

Increasing age and female sex are well-known risk factors for knee OA in all regions [13]. Our findings are in agreement: women had a 2.1-fold greater risk of radiographic knee OA than did men, and persons aged $\geq 80$ years had an approximately 9-fold greater risk of radiographic knee OA than did those aged 50–59 years.

Rural residence was also a risk factor for radiographic knee OA in this study, even after adjustment for age, sex, education level, number of family members, household income, occupation, and obesity.

Population-based surveys conducted in urban Beijing and rural Wuchuan County, China, also showed that men and women in Wuchuan had roughly double the prevalence of knee OA compared with their Beijing counterparts [14]. In elderly Japanese population-based cohorts, residents of mountainous areas had a greater risk of radiographic knee OA (Kellgren–Lawrence grade $\geq 2$) than did urban residents, indicating the involvement of environmental factors such as nutrition or occupation (e.g., farming, forestry), which demands physical activity and repetitive laborious use of the knee joints [15].

In this study, the weighted prevalence of obesity, defined as BMI $\geq 25$ kg/m$^2$, was approximately 35%. Obesity was associated with an approximately two-fold greater risk of knee OA, regardless of knee OA severity (Kellgren–Lawrence grades $\geq 2$, $\geq 3$, and 4). This result was similar to the findings of a meta-analysis, which yielded a pooled OR of 2.1 (95% CI 1.82–2.42), indicating an increased risk of knee OA, in overweight (BMI 25–30 kg/m$^2$) and obese (BMI > 30 kg/m$^2$) individuals [13]. Obesity plays a role in the development and progression of knee OA through variable combinations of mechanical, humeral, and metabolic factors, including elevated adipocytokine levels and associated pro-inflammatory responses, as well as mechanical loading of the knee joint during weight bearing [16].

We found that a low education level and low household income were associated with radiographic knee OA. A few studies have revealed associations between low socioeconomic status and knee OA [17–20]. Callahan *et al.* reported that low educational attainment, but not occupation, was associated significantly with radiographic knee OA.[18]. According to the China Health and Retirement Longitudinal Study, knee OA was more prevalent in subjects who had received less education than in those who had received more education [21]. A low education level could lead to reduced health literacy and health-promoting activities. Jorgensen *et al.* suggested that lifestyle differences are responsible, at least in part, for the reduced risk of knee OA in persons with more education and higher than average incomes, based on finding from their study of a Danish cohort [22].

Occupational activity, which includes kneeling, squatting, lifting, and climbing stairs at work, is a modifiable risk factor for the development and progression of knee OA [23]. One study showed that male farmers, construction workers, and firefighters had increased risks of knee OA [24]. Based on 2010–2012 KNHANES data, Kim et al. reported that male low-level workers (skilled agricultural and fishery workers) and blue-collar workers (technicians and device and machine operators) aged $\geq 50$ years were at greater risk of knee OA and chronic knee pain [25]. In our study, weighted prevalences of radiographic knee OA were higher in skilled agricultural, forestry, and fishery workers and in craft, plant, or machine operators and assemblers compared with managers and professionals, after adjustment for age and sex. However, after adjustment for age, sex, area of residence, education level, number of family members, household income, and obesity, the statistical significance did not persist.

This study has several strengths. First, we examined a large, nationally representative sample of adult Koreans. To our knowledge, few other studies have been based on national surveillance of knee OA in the general population that included >10,000 subjects using sampling

weights. Second, we excluded subjective self-reported knee pain and focused on the radiographic findings of knee OA. The knee OA grading agreement rate was high, and coefficients of inter-rater reliability between radiologists showed fair to very high degrees of agreement. Third, we identified sociodemographic factors associated with radiographic knee OA from nation-representative sample in Korea. An enhanced understanding of the demographic factors associated with knee OA provides information on the population at high risk of knee OA, which is useful for prevention and management in the early stages.

Nevertheless, our study has some limitations. First, radiographic findings of knee OA are usually, but not always, correlated with patient symptoms; radiographic OA changes are not always associated with knee pain [26,27]. As we did not consider knee pain, the prevalence of knee OA may have been over- or underestimated in this study. Second, although we adjusted for many covariates, the effects of residual or hidden confounding variables cannot be excluded, similar to other cross-sectional studies.

In conclusion, using data from the 2010–2013 KNHANES and defining knee OA as Kellgren–Lawrence grade $\geq 2$, we found that the prevalence of radiographic knee OA was 35.1% (24.4% in men, 44.3% in women) in a representative sample of Korean adults aged $\geq 50$ years, with the highest prevalence (78.7%) observed in women aged $\geq 80$ years. Low socioeconomic status and traditional factors, including age, female sex, and obesity, were associated with the risk of knee OA. To reduce inequalities in knee OA prevalence, interventions and policies should target low-socioeconomic-status groups.

## Author Contributions

**Conceptualization:** Dong-Jun Kim.

**Data curation:** Dong-Jun Kim.

**Formal analysis:** Dong-Jun Kim.

**Methodology:** Dong-Jun Kim.

**Writing – original draft:** Jae Won Hong.

**Writing – review & editing:** Jung Hyun Noh, Dong-Jun Kim.

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
