## [Decision Letter · Decision Letter 0]

10 Oct 2019

PONE-D-19-15672

The Prevalence of and Demographic Factors Associated with Radiographic Knee Osteoarthritis in Korean Adults Aged ≥ 50 years: The 2010–2013 Korea National Health and Nutrition Examination Survey

PLOS ONE

Dear Dr. Kim,

Thank you for submitting your manuscript to PLOS ONE. After careful consideration, we feel that it has merit but does not fully meet PLOS ONE’s publication criteria as it currently stands. Therefore, we invite you to submit a revised version of the manuscript that addresses the points raised during the review process.

We would appreciate receiving your revised manuscript by Nov 24 2019 11:59PM. To enhance the reproducibility of your results, we recommend that if applicable you deposit your laboratory protocols in protocols.io, where a protocol can be assigned its own identifier (DOI) such that it can be cited independently in the future. For instructions see: http://journals.plos.org/plosone/s/submission-guidelines#loc-laboratory-protocols

We look forward to receiving your revised manuscript.

Kind regards,

Young Dae Kwon, M.D., Ph.D.

Academic Editor

PLOS ONE

Journal Requirements:

Reviewers' comments:

Reviewer's Responses to Questions

**Comments to the Author**

1. Is the manuscript technically sound, and do the data support the conclusions?

Reviewer #1: No

2. Has the statistical analysis been performed appropriately and rigorously? 

Reviewer #1: No

3. Have the authors made all data underlying the findings in their manuscript fully available?

Reviewer #1: Yes

4. Is the manuscript presented in an intelligible fashion and written in standard English?

Reviewer #1: No

5. Review Comments to the Author

Reviewer #1: Please see my Major comments below:

1. This reader does not agree with the authors’ statement “Evidence for the sociodemographic factors affecting radiographic knee OA, other than traditional factors such as age, female sex, and obesity, is limited.”

a. First, this statement is vague in providing the rationale for this current study.

b. Second, many of the factors in the current analysis have been examined previously in NHANES in the US, as well as studies in China, and other countries worldwide. For example, factors such as income, education, occupation, geographical area of residence in addition to the more “traditional” factors such as sex, age, and BMI have been examined as evidenced by the authors citing of previous work in their Discussion section. In fact, many factors were also examined in the 2009 Korean NHANES by Lee KM et al. Yonsei Med J 56(1):124-131, 2015.

c. Thus, the novelty of and rationale for this study need to be better defined.

2. Methods section is not clearly written with respect to statistical analysis and variables of interest.

a. First, it is unclear the methods used to account for sampling weights?

b. Second, please clarify the statements “To compare the weighted prevalence of radiographic knee OA by sociodemographic factors, chi-squared tests and analysis of covariance were performed. The prevalence of radiographic knee OA was analyzed using age, sex, area of residence, education level, number of family members, household income, occupation, and obesity [body mass index (BMI) ��25 kg/m2] as confounding variables.”

I may be mistaken, but my understanding of analysis of covariance (ANCOVA) is that it is used to test the main and possible interaction effects of categorical variables on a continuous dependent variable, controlling for other continuous variables. It is unclear what this main dependent variable is? Was Radiographic OA as measured with KL grade used as a continuous/ordinal variable in order to justify use of ANCOVA? If so, this cannot be a measure of prevalence? Therefore, it is unclear how prevalence of radiographic knee OA (which I am assuming is a proportion) was defined. If proportion of those with KL >=2 vs. those <2; or >=3 vs those <3; or those =4 vs. <4, was defined and what analyses were used to adjust for “confounders” in the prevalence of ROA?

c. My understanding of a confounding variable is that it is an “extraneous” risk factor for ROA other than the exposure of interest and needed to be controlled for in the analysis. We usually control for confounding in relation to an exposure of interest. It seems to me that age, sex, area of residence, education, # family members, household income, occupation, BMI >= 25, are not necessarily confounding variables in this research project, and that they are used more like predictors of, or covariates in a prediction model with ROA (defined one of 3 ways: as KL >=2 vs. <2, >=3 vs. <3, or =4 vs. <4) as the outcome?

3. Methods and Tables should stand alone? In other words, a reader should not have to study the tables to understand how variables of interest were defined in a study. For example, I believe that categories of age, education, income, occupation, etc.. should be defined in the Methods section. Also, how was rural vs. urban defined?

4. Results section:

a. Please clarify what “irrespective of severity” means.

b. As for the Tables such as Table 3, what does “other variables” mean? Is Table 4 based on multivariable logistic regression? If so, need to mention these variables are mutually adjusted? is used in the models. Is it used as a continuous variable or as categorical variable? How was rural vs. urban defined?

c. Given separate models for different definitions of ROA and separate models where different sets of covariates were used, it would be clearer to understand what the authors mean by prevalence(s) of radiographic knee OA or “after adjustment” in the Results.

d. Since prevalence and incidence of OA are very different in men compared with women, and factors associated with OA may be very different in the two groups, we tend to stratify by men and women in analyses? Why did the authors NOT conduct the analyses stratified by men and women? Would these factors, whether sociodemographic or lifestyle factors differ in men and women, including that of occupation?

5. In the ABSTRACT, the last sentence in the Results section related to the use of Logistic regression analyses should be under the Methods section? Again, because the authors mentioned that ROA was defined as KL >=2 vs. < 2, then several times mentioned “regardless of OA severity”—this was a bit confusing because data for “severity” were not presented in abstract, as the Abstract is a stand-alone document I read before I read the results in the main paper.

6. PLOS authors have the option to publish the peer review history of their article (what does this mean?). If published, this will include your full peer review and any attached files.

Reviewer #1: No

---

## [Author Response · Author response to Decision Letter 0]

21 Nov 2019

Reviewer #1: Please see my Major comments below:

1. This reader does not agree with the authors’ statement “Evidence for the sociodemographic factors affecting radiographic knee OA, other than traditional factors such as age, female sex, and obesity, is limited.”

a. First, this statement is vague in providing the rationale for this current study.

b. Second, many of the factors in the current analysis have been examined previously in NHANES in the US, as well as studies in China, and other countries worldwide. For example, factors such as income, education, occupation, geographical area of residence in addition to the more “traditional” factors such as sex, age, and BMI have been examined as evidenced by the authors citing of previous work in their Discussion section. In fact, many factors were also examined in the 2009 Korean NHANES by Lee KM et al. Yonsei Med J 56(1):124-131, 2015.

c. Thus, the novelty of and rationale for this study need to be better defined.

Thank you so much for your valuable comments.

a. We totally agree with you. We deleted the sentences you mentioned (“Evidence for the sociodemographic factors affecting radiographic knee OA, other than traditional factors such as age, female sex, and obesity, is limited.”) in abstract and introduction section.

b. We fully understand your concern. However, there are some differences between previous and our study in Korea.

Lee KM et al. used data from 1,728 subjects aged > 65 years from the overall dataset and reported risk factors for OA (unspecified site) based on self-reports. And they investigated the association in 1,728 subjects, not in weighted-sample. On the other hand, we excluded subjective self-reported knee pain and focused on the radiographic findings of knee OA. Our study used data from 12,287 individuals aged ≥ 50 years (weighted number of subjects, 14,837,279 persons) and demonstrated the weighted prevalence and weighted associated factors of knee OA based on radiographic finding. 

c. We understand your concern about the novelty of and rationale for this study. However, to our knowledge, few epidemiologic studies have been based on national surveillance of knee OA in the general population that included >10,000 subjects using sampling weights. The most important meaning of this study will be the presentation of radiographic finding-verified knee OA prevalence in nation-representative samples of Korea, because the prevalence of knee OA could be different among nations and the presentation of knee OA prevalence in whole population of one country could be useful for further understanding and future studies in the field of OA. Although there is no novel associated factor with knee OA in this study compared to previous studies, the investigation of demographic factors associated with the presence of knee OA could be meaningful because the results of this study were drawn from nation-representative sample in Korea. 

2. Methods section is not clearly written with respect to statistical analysis and variables of interest.

a. First, it is unclear the methods used to account for sampling weights?

Sampling weights were provided by Korea CDC (Center for Disease Control) when we download raw data from the Korea National Health and Nutrition Examination Survey database website. Details of methods used to account for sampling weights are presented in ‘the guideline for analyses of Korea NHANES dataset by Korea CDC (in Korean)’, which can be downloaded from the Korea NHNES database website.

b. Second, please clarify the statements “To compare the weighted prevalence of radiographic knee OA by sociodemographic factors, chi-squared tests and analysis of covariance were performed. The prevalence of radiographic knee OA was analyzed using age, sex, area of residence, education level, number of family members, household income, occupation, and obesity [body mass index (BMI) ��25 kg/m2] as confounding variables.”

I may be mistaken, but my understanding of analysis of covariance (ANCOVA) is that it is used to test the main and possible interaction effects of categorical variables on a continuous dependent variable, controlling for other continuous variables. It is unclear what this main dependent variable is? Was Radiographic OA as measured with KL grade used as a continuous/ordinal variable in order to justify use of ANCOVA? If so, this cannot be a measure of prevalence? Therefore, it is unclear how prevalence of radiographic knee OA (which I am assuming is a proportion) was defined. If proportion of those with KL >=2 vs. those <2; or >=3 vs those <3; or those =4 vs. <4, was defined and what analyses were used to adjust for “confounders” in the prevalence of ROA?

-> We are very sorry for our terrible mistakes. As you pointed out, ANCOVA is an analysis for continuous variable as an outcome. We analyzed by using GLM (general linear model), not ANCOVA, and corrected mistake in method section. We used GLM analysis of complex analysis in SPSS. 

“To compare the weighted prevalence of radiographic knee OA by sociodemographic factors, chi-squared tests and general linear model were performed.”

c. My understanding of a confounding variable is that it is an “extraneous” risk factor for ROA other than the exposure of interest and needed to be controlled for in the analysis. We usually control for confounding in relation to an exposure of interest. It seems to me that age, sex, area of residence, education, # family members, household income, occupation, BMI >= 25, are not necessarily confounding variables in this research project, and that they are used more like predictors of, or covariates in a prediction model with ROA (defined one of 3 ways: as KL >=2 vs. <2, >=3 vs. <3, or =4 vs. <4) as the outcome?

Thank you for your comment. We revised “the confounding variables” to “covariates” in abstract, method, and discussion section.

3. Methods and Tables should stand alone? In other words, a reader should not have to study the tables to understand how variables of interest were defined in a study. For example, I believe that categories of age, education, income, occupation, etc.. should be defined in the Methods section. Also, how was rural vs. urban defined?

We modified all the following sentences as your recommendation.

“ The prevalence of radiographic knee OA was analyzed using age (50-59, 60-69, 70-79, � 80 years old), sex (men/women), area of residence (urban[Dong]/rural[Eup/Myeon), education level (elementary school/junior high school/senior high school/college graduated), number of family members (1/2/3/�4), household income (� 24th, 25-49th, 50-74th, ≥75th percentile), occupation (managers and professionals/clerical support workers/service and sales workers/skilled agricultural, forestry and fishery workers/craft, plant, or machine operators and assemblers/laborers/unemployed (including students and house wives), and obesity [body mass index (BMI) ≥ 25 kg/m2] as covariates.”

According to administrative district of South Korea, “dong” is classified into the urban area, and “eup” / “myeon” are classified into to the rural area.

We added these administrative districts in Method section.

“area of residence (urban[Dong]/rural[Eup/Myeon)”

4. Results section:

a. Please clarify what “irrespective of severity” means.

We clarified the “severity” to “knee OA severity (Kellgren–Lawrence grades ≥2, ≥3, and 4)”

b. As for the Tables such as Table 3, what does “other variables” mean? Is Table 4 based on multivariable logistic regression? If so, need to mention these variables are mutually adjusted? is used in the models. Is it used as a continuous variable or as categorical variable? How was rural vs. urban defined?

We added specific variables “area of residence, education level, number of family members, household income, occupation, and obesity.”, instead of “other variables” in Result section and Table 3.

Table 4 is analyzed based on multivariable logistic regression as categorical variable. We added the categorical definition at each variables in Method section.

“Logistic regression analyses were used to calculate the odds ratio (OR) for radiographic knee OA with age(50-59, 60-69, 70-79, ≥80 years old), sex(men/women), area of residence(urban[Dong]/rural[Eup/Myeon), education level(elementary school/junior high school/senior high school/college graduated), household income(�24th, 25-49th, 50-74th,≥75th percentile), and obesity(no/yes) serving as covariates.”

According to administrative district of South Korea, “dong” is classified into the urban area, and “eup” / “myeon” are classified into to the rural area.

We added these administrative districts in Method section.

c. Given separate models for different definitions of ROA and separate models where different sets of covariates were used, it would be clearer to understand what the authors mean by prevalence(s) of radiographic knee OA or “after adjustment” in the Results.

We revised the results from unadjusted to adjusted prevalence of radiographic knee OA

“The unadjusted and adjusted weighted prevalences of radiographic knee OA were lower in urban areas than in rural areas [adjusted, 34.0% (32.6–35.5%) vs. 38.2% (35.5–40.9%), p = 0.008].

Education level was correlated inversely with the prevalence of radiographic knee OA, before and after adjustment. Elementary school graduates had a higher prevalence of radiographic knee OA than did college graduates [adjusted, 37.6% (35.7–39.5%) vs. 29.5% (26.7–32.3%), p < 0.001].”

d. Since prevalence and incidence of OA are very different in men compared with women, and factors associated with OA may be very different in the two groups, we tend to stratify by men and women in analyses? Why did the authors NOT conduct the analyses stratified by men and women? Would these factors, whether sociodemographic or lifestyle factors differ in men and women, including that of occupation?

We agree with your opinion.

We added table 5 which showed logistic regression analyses for radiographic knee osteoarthritis (Kellgren-Lawrence grade ≥ 2) according to sex.

5. In the ABSTRACT, the last sentence in the Results section related to the use of Logistic regression analyses should be under the Methods section? Again, because the authors mentioned that ROA was defined as KL >=2 vs. < 2, then several times mentioned “regardless of OA severity”—this was a bit confusing because data for “severity” were not presented in abstract, as the Abstract is a stand-alone document I read before I read the results in the main paper.

To avoid confusion, we added the definition of Kellgren–Lawrence grade in methods section in the Abstract.

---

## [Editor Report · Decision Letter 1]

5 Mar 2020

The Prevalence of and Demographic Factors Associated with Radiographic Knee Osteoarthritis in Korean Adults Aged ≥ 50 years: The 2010–2013 Korea National Health and Nutrition Examination Survey

PONE-D-19-15672R1

Dear Dr. Kim,

We are pleased to inform you that your manuscript has been judged scientifically suitable for publication and will be formally accepted for publication once it complies with all outstanding technical requirements.

With kind regards,

Young Dae Kwon, M.D., Ph.D.

Academic Editor

PLOS ONE

---

## [Editor Report · Acceptance letter]

9 Mar 2020

PONE-D-19-15672R1 

The Prevalence of and Demographic Factors Associated with Radiographic Knee Osteoarthritis in Korean Adults Aged ≥ 50 years: The 2010–2013 Korea National Health and Nutrition Examination Survey 

Dear Dr. Kim:

I am pleased to inform you that your manuscript has been deemed suitable for publication in PLOS ONE. Congratulations! Your manuscript is now with our production department. 

With kind regards,

on behalf of

Dr. Young Dae Kwon 

Academic Editor

PLOS ONE